# Influence of the representation of convection on the mid-Holocene West African Monsoon

Leonore Jungandreas[1], Cathy Hohenegger[1], and Martin Claussen[1,3]

[1]Max Planck Institute for Meteorology, Bundesstraße 53, 20146 Hamburg
[3]Center for Earth System Research and Sustainability, Universität Hamburg, Bundesstraße 53, 20146 Hamburg

**Correspondence:** Leonore Jungandreas (leonore.jungandreas@mpimet.mpg.de)

**Abstract.** Global climate models have difficulties to simulate the northward extension of the monsoonal precipitation over north Africa during the mid-Holocene as revealed by proxy data. A common feature of these models is that they usually operate on too coarse grids to explicitly resolve convection, but convection is the most essential mechanism leading to precipitation in the west African monsoon region. Here, we investigate how the representation of tropical deep convection in the ICON climate model affects the meridional distribution of monsoonal precipitation during the mid-Holocene, by comparing regional simulations of the summer monsoon season (July to September, JAS) with parameterized (40 km-P) and explicitly resolved convection (5 km-E).

In the 5 km-E simulation, the more localized nature of precipitation and the absence of permanent light precipitation as compared to the 40 km-P simulation is closer to expectations. However, in the JAS-mean the 40 km-P simulation produces more precipitation and extents further north than the 5 km-E simulation, especially between $12\,^\circ$ N and $17\,^\circ$ N. The higher precipitation rates in the 40 km-P simulation are consistent with a stronger monsoonal circulation over land. Furthermore, the atmosphere in the 40 km-P simulation is less stably stratified and notably moister. The differences in atmospheric water vapor are the result of substantial differences in the probability distribution function of precipitation and its resulting interactions with the land surface. The parametrization of convection produces light and large-scale precipitation, keeping the soils moist and supporting the development of convection. In contrast, less frequent but locally intense precipitation events lead to high amounts of runoff in the explicitly resolved convection simulations. The stronger runoff inhibits the moistening of the soil during the monsoon season and limits the amount of water available to evaporation in the 5 km-E simulation.

## 1  Introduction

During the mid-Holocene, around 9000 to 6000 years before present (yBP), the landscape of the today's extremely arid Sahara was transformed into a widespread savannah-like landscape characterized by grass- and shrublands (Jolly et al., 1998), variable tree cover and permanent lakes and wetlands (Tierney et al., 2017). This remarkable transformation of the Sahara, which is commonly called "Green Sahara", can be attributed to an intensified West African monsoon (WAM) (Kutzbach and Otto-

Bliesner, 1982; Kutzbach and Liu, 1997). The intensification of the WAM circulation was driven by a higher summer insolation in the Northern Hemipshere during the mid-Holocene (Kutzbach and Guetter, 1986; Street-Perrott et al., 1990). Reconstructions of precipitation from proxy data (Peyron et al., 2006; Bartlein et al., 2011) indicate around 200 to 700 mm year$^{-1}$ more precipitation over the Sahel-Saharan region during this humid period. However, global General Circulation Models (GCMs) neither capture the reconstructed mean precipitation (Yu and Harrison, 1996; Braconnot et al., 2012) nor the change in precipitation between the coastal regions of Africa and the arid Sahel-Sahara for the mid-Holocene (Joussaume et al., 1999; Braconnot et al., 2012; Harrison et al., 2015; Brierley, 2020). Compared to reconstructions most climate models (e.g. used in PMIP3 and PMIP4) are able to reproduce the precipitation amount over the Sahel but produce too little precipitation north of 15 °N, resulting in an overly strong meridional precipitation gradient .

The reasons for the mismatch between climate models and reconstructions are still debated. Feedback mechanisms between the land/vegetation (e.g. Kutzbach and Liu (1997); Claussen and Gayler (1997); Braconnot et al. (1999, 2012); Claussen et al. (2017)), ocean (e.g. Kutzbach and Liu (1997); Hewitt and Mitchell (1998); Braconnot et al. (1999, 2012)) and the atmosphere are known to enhance the orbitally induced increase in mid-Holocene monsoonal precipitation (Joussaume et al., 1999; Braconnot et al., 2012). A better representation of the land surface in GCMs may be necessary to substantially increase precipitation levels (e.g. Levis et al. (2004); Vamborg et al. (2010)). As another factor, changes in dust fluxes between present-day and mid-Holocene conditions have been mentioned and are a subject of controversy (Pausata et al., 2016; Thompson et al., 2019). Finally, by adding an artificial heating source within the atmospheric boundary layer over the Sahara, Dixit et al. (2018) were able to increase the magnitude and northward extent of precipitation comparable to what is seen in proxy data. They argued that GCMs miss an important local diabatic heating source over the Sahel-Saharan region.

The parameterization of convection poses another limitation for GCMs. For most, if not all, palaeo simulations, GCMs usually operate on relatively coarse horizontal resolution ($\sim$ 200 km), where convection is not explicitly resolved. Several studies (Yang and Slingo, 2001; Randall et al., 2003; Stephens et al., 2010; Dirmeyer et al., 2012; Fiedler et al., 2020) have shown that simulations with parameterized convection are not able to reproduce many key characteristics of the present-day precipitation distribution, such as the location of the ITCZ, the propagation of the monsoon or the diurnal cycle of precipitation. Also, they produce too much and too light rainfall.

This raises the question as to whether convection-permitting simulations can improve the representation of the WAM and the precipitation distribution for mid-Holocene climate conditions. Support for this hypothesis comes from the study by Marsham et al. (2013). For present-day conditions, they conducted short (covering only 10-days) simulations with explicitly resolved convection and parametrized convection for a regional domain located in northwest Africa. In their study, the monsoonal precipitation propagated further northward and peaked between 10 ° N to 12 ° N in their simulation with explicit convection, in better agreement with observations. They ascribed the improvement of the precipitation pattern to the better representation of the diurnal cycle of convection.

Using the ICON-NWP model (ICOsahedral Nonhydrostatic model framework for Numerical Weather Prediction), we investigate how the representation of convection impacts the mid-Holocene WAM. We perform parameterized and explicitly resolved convection simulations with prescribed mid-Holocene atmospheric initial and boundary conditions for two entire monsoon

seasons. The main aim of the study is to test whether explicitly resolving convection leads to a stronger northward propagation
of the WAM during the mid-Holocene.

The paper is structured as follows: We describe the model and different simulation setups in section 2. In section 3, we present
and explain the simulated precipitation patterns. A summary and conclusion follows in section 4.

## 2 Methods

### 2.1 Model

We use the ICON (ICOsahedral Nonhydrostatic) model framework version 2.5.0 (Zängl et al., 2015) in its operational Nu-
merical Weather Prediction (NWP) mode. ICON was developed through a collaboration between the Max-Planck Institute
for Meteorology and the German Weather Service. The model has already been used and evaluated with respect to tropical
convection and circulation by several studies, (e.g. Klocke et al. (2017); Stevens et al. (2019); Hohenegger et al. (2020)). Zängl
et al. (2015) lists the physical parameterizations of the model framework. The parameterization of convection is based on
the bulk mass-flux approach introduced by Tiedtke (1989) with modifications by Bechtold et al. (2014). We will switch the
convective parameterization on or off, depending on the grid spacing. Our limited-area simulations are forced with initial and
boundary data from a transient global Holocene simulation, previously conducted with the MPI - Earth System Model (ESM)
and covering the years from 8000 BP (before present where year 2000 A.D. is used as reference) to 150 BP. Dallmeyer et al.
(2020) describes the performance of the transient MPI-ESM Holocene simulations in detail. Furthermore, we prescribe 6 -
hourly sea surface temperature (SST) and sea ice (SIC) fields originating from the transient Holocene simulations. The orbital
parameters and the tracer gases, carbon dioxide ($CO_2$), methane ($CH_4$) and nitrogen oxide ($N_2O$), reflect mid-Holocene con-
ditions, as in the MPI-ESM Holocene simulation. Concerning the description of the land surface and vegetation, we take the
external parameters from reanalysis data of the Integrated Forecast System (IFS) of the European Centre for Medium-Range
Weather Forecasts (ECMWF). Similar to the setup of the simulations in the first Palaeoclimate Modelling Intercomparison
Project (PMIP; i.e Joussaume et al. (1999); Braconnot et al. (2000, 2004)), the external data reflect present-day conditions. By
using present-day surface conditions in this study we, for a first step, ignore the enhancement of precipitation due to extented
vegetation and wetter soil moisture conditions. Furthermore, with a higher vegetation cover the atmosphere-soil hydrology
interactions are also likely to change substantially. Plants enhance the infiltration of water into the soil and increase the inter-
ception storage of water. Therefore, they increase evapotranspiration into the atmosphere from interception and by transpiration
of water from deeper soil layers.

### 2.2 Simulation Setup

Firstly, we perform a 30 - year "spinup" simulation on a regional domain (see Fig. 1) with 40 km horizontal grid spacing and 75
vertical levels. The spinup simulation runs for the period 7039 BP to 7010 BP. The convective parameterization is active in this
simulation. After around 15 years the soil moisture equilibrates to a stable state. We choose two years after the 15-year spinup

phase and perform several nesting experiments for the boreal summer monsoon season. The nesting experiments are initialized for 30 th May and run for five months (JJASO). The parent domain of the nested simulation is identical to the domain of the spinup simulation with the same horizontal and vertical resolution. The nesting configuration then reduces the horizontal grid spacing by a factor of two down to the 5 km horizontal resolution (Fig. 1). The nested simulations with 40 km, 20 km and 10 km grid spacing are run with parameterized convection. In the following we refer to these simulations as the 40 km-P, the 20 km-P and the 10 km-P simulations. The 5 km simulations resolve convection explicitly and are referred to as the 5 km-E simulations. We simulate with a one-way nesting strategy. The nested simulations are initalized one hour after another. Lateral boundary conditions for the nested simulations are obtained from their parent simulation and updated every 6 hours.

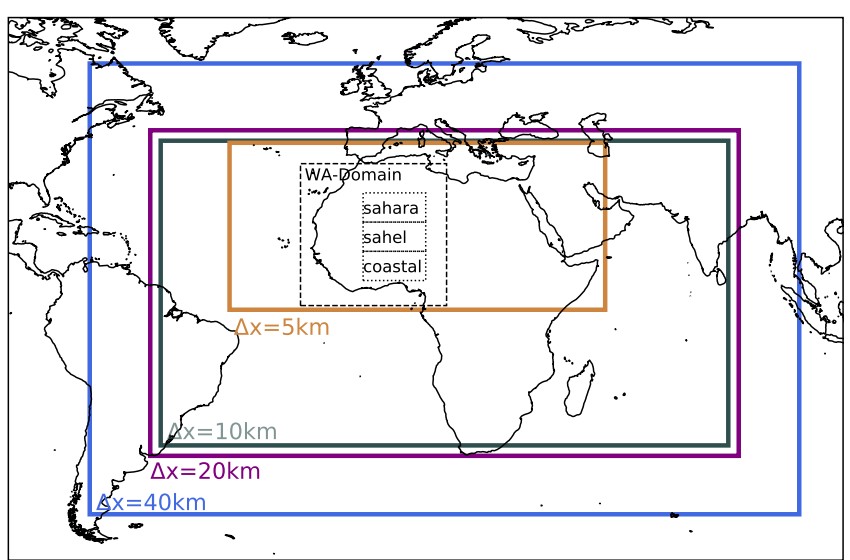

**Figure 1.** Coloured domains display the nesting domains of the simulations for the various grid spacings (as indicated). The dashed, black domain with the label "WA-Domain" (West Africa) shows the anaylsis domain for Fig. 4 and Fig. 10 and spans the area from 1 °N-35 °N and from 20 °E-15 °W. The three dotted, black domains over north Africa are used to calculate the skew - T diagrams for the coastal african region ("coastal"), the sahel region ("sahel") and the saharan region ("sahara") in Fig. 6. The coastal region spans from 6 °N-14 °N and from 5 °E-10 °W. The sahel region spans from 14 °N-21 °N and from 5 °E-10 °W and the sahara region from 21 °N-28 °N and from 5 °E-10 °W.

For each nesting suite, we perform two simulations, one for the year 7023 BP and one for the year 7019 BP. We selected these two years from the spinup simulations based on the simulated mean precipitation (Fig. 2 a) and the mean meridional distribution of precipitation (Fig. 2 b) within each year from June to October and over land points of the 5 km-domain (Fig. 1). We chose the year 7023 BP as it displays a combination of generally higher precipitation amounts and slightly higher precipitation rates at latitudes north of 15 ° N relative to the other simulated spinup years. The year 7019 BP, in contrast, gives the combination

of slightly weaker precipitation amounts and weaker precipitation rates north of 15 ° N compared to most of the other years of the spinup simulation.

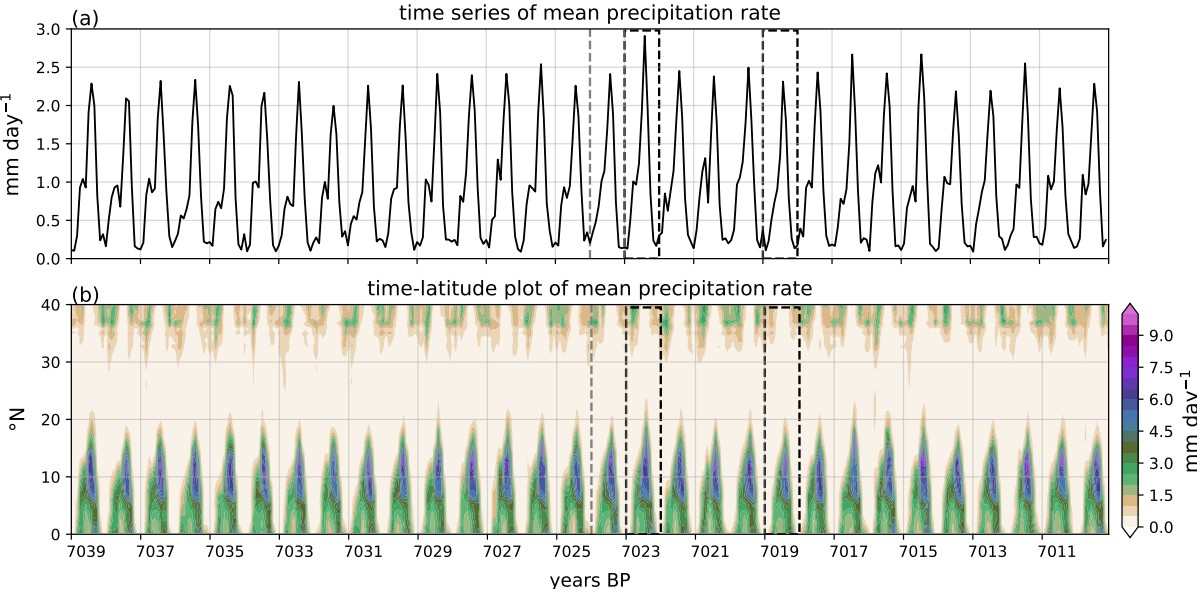

**Figure 2.** 30-year timeseries of the mean precipitation rate (a) and the latitude-time plot of the mean precipitation rate over land (b) indicating the northward extension of the WAM every year for the whole 5 km domain spanning the area from 0 °N-40 °N and from 37 °E-53 °W. The dashed,light-grey line indicates the end of the spinup-phase when soil moisture is on a constant level. The dashed rectangles show the years we chose for our analysis: 7023 BP (strong) and 7019 BP (weak).

Marsham et al. (2013) identified the difference in the simulated precipitation diurnal cycle between explicitly resolved and parameterized convection as the main reason for the different meridional distributions of precipitation for his simulations under present-day conditions. In contrast to Marsham et al. (2013) and to GCMs used in PMIP, the convective parameterization used in the operational setup of ICON-NWP simulates the peak of diurnal convection lin the late afternoon instead of noon in agreement with observations, due to modifications introduced by Bechtold et al. (2014). Thus, the timing of the precipitation diurnal cycle in simulations with explicit and parameterized convection is similar in our case (Figure 3 a). To test the importance of the timing of the diurnal cycle of precipitation for the representation of the monsoon propagation during the mid-Holocene, we perform a second set of nested simulations where we removed the modifications by Bechtold et al. (2014). In this simulation with modified diurnal cycle, the convection peaks around noon, as expected (Fig. 3b). Even though the diurnal cycle of the 5 km-E simulation is not directly modified in this second set of nested simulations, comparing this simulation to our control simulation reveals small differences as the two 5 km-E simulations are driven by distinct 40 km-P simulations. In the following, simulations with the modified diurnal cycle are labeled with "mod" (Sec. 3.6).

We perfom a third suite of nested simulations to separate the impact of resolution on the WAM from the impact of the representation of convection. In these simulations the 20 km and 10 km simulations are run with explicitly resolved convection. These
120 simulations are referred to as the 20 km-E and the 10 km-E simulation. We compare these with the 20 km-P and the 10 km-P simulation.

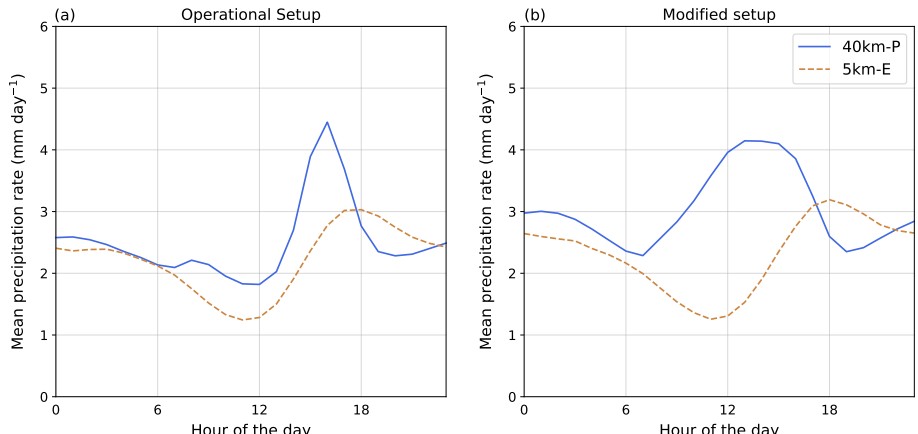

**Figure 3.** Mean diurnal cycle of precipitation for the 40 km-P (blue-soild) and the 5 km-E (orange-dashed) simulation for JAS. On the left, the diurnal cycle for the operational ICON-NWP model; on the right, the diurnal cycle of the modified ICON-NWP setup used in Sec. 3.6. The diurnal cycle is calculated over the dashed "WA-domain" outlined in Fig. 1.

## 3 Results and Discussion

### 3.1 Precipitation distribution

During the mid-Holocene, the northward propagation of precipitation constitutes the main difference to today's precipitation
pattern. As described in the introduction, reconstructions point towards less precipitation over equatorial Africa and the Sahel region but substantially more precipitation over the Sahara. Therefore, we are mainly interested in the meridional precipitation gradient which modulate the landscape and vegetation cover of the north African continent.

In the JJASO simulations with ICON-NWP we identify July to September (JAS) as the strongest monsoon months. Therefore we mainly focus our analysis on these three months. Furthermore, as the simulations for the two years (7023 BP and 7019 BP)
reveal similar results, we only show the results for 7023 BP.

Figure 4 a clearly shows that the 40 km-P simulation produces more precipitation and precipitation that reaches further north than the 5 km-E simulation. On average and from $12\,^{\circ}$ N to $17\,^{\circ}$ N, it rains 0.8 mm day$^{-1}$ more in the 40 km-P simulation than in the 5 km-E simulation at each latitude. Over five months this sums up to over 120 mm per latitude which could already impact the vegetation cover. A secondary precipitation peak is also visible around $15\,^{\circ}$ N in the 40 km-P simulation, a peak that

is absent in the 5 km-E simulation. The fact that the parameterized simulation produces more precipitation is not only true for the 40 km-P simulation but also for all grid spacings where parameterized convection is used.

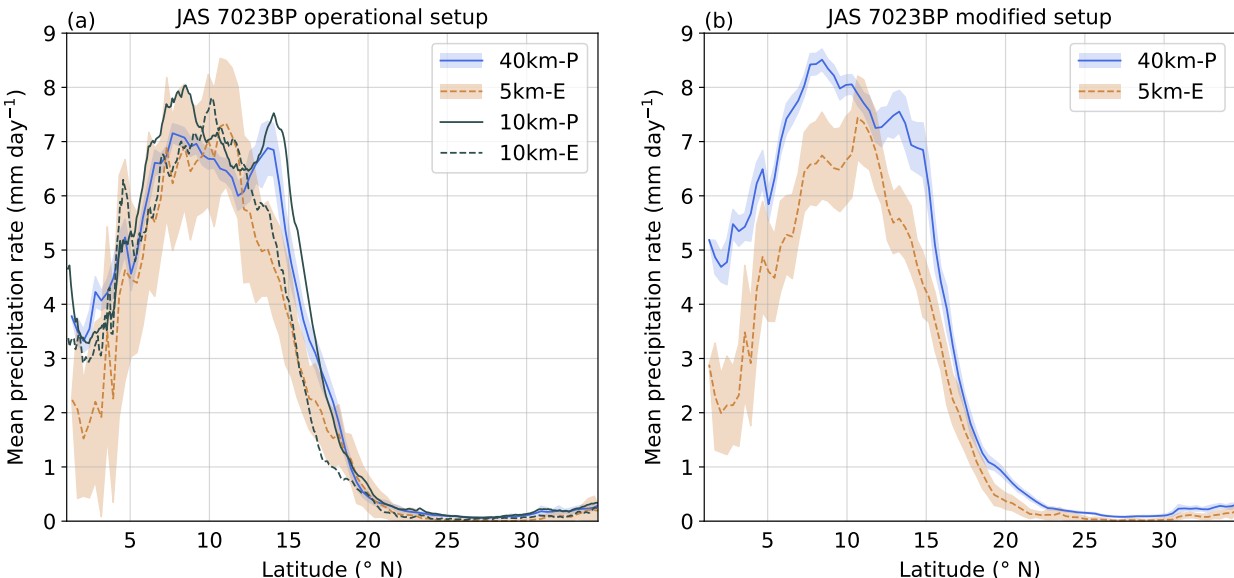

**Figure 4.** Meridional distribution of monsoonal precipitation for a) the operational simulations and b) the setup with modified diurnal cycle (analyzed in Sect. 3.6) for the 40 km-P (blue-solid), the 5 km-E (orange-dashed), the 10 km-P (black-solid) and the 10 km-E (black-dashed) simulation. The shading displays the daily longitudinal mean standard deviation from the longitudinal mean JAS cycle for the 40 km-P and the 5 km-E simulation. The lonitudinal mean is taken over the land points of the WA-domain outlined in Fig. 1.

Figure 4 a also shows that the mean meridional distribution of monsoonal precipitation is predominantly determined by the representation of convection rather than the resolution. This becomes visible by comparing the 10 km-P and the 10 km-E

simulation. The 10 km-P simulation is more similar to the 40 km-P simulation than to the 10 km-E simulation. Conversely, the 10 km-E simulation is more similar to the 5 km-E simulation than to 10 km-P. This is also valid for the 20 km-P and 20 km-E simulations (not shown). Given these similarities and the fact that grid spacings finer than 5 km are typically employed when conducting convection-permitting simulations, we focus our analysis to th 40 km-P and 5 km-E simulation. The analysis was repeated for the 10 km-P and 10 km-E simulation and can be found in the Appendix A.

In the following, we begin by analyzing the large-scale mean state of the atmosphere by examining the pressure field and the large-scale dynamics of the WAM circulation to understand the precipitation differences (Sect. 3.2). We then investigate whether and how the thermodynamic structure of the atmosphere supports the development of convection and precipitation in the 40 km-P and 5 km-E simulations (Sect. 3.3). As this analysis points to strong differences in the moisture field between the 40 km-P and the 5 km-E simulation, we examine in the follwoing two sections differences in moisture transport (Sect. 3.4) and

differences in evapotranspiration (Sect. 3.5), the two moisture sources for precipitation. Finally, we test whether the diurnal cycle of convection impacts the propagation of the WAM over north Africa as suggested by Marsham et al. (2013) (Sect. 3.6).

## 3.2   Large - Scale Circulation

To understand the unexpectedly stronger precipitation in the 40 km-P simulation compared to the 5 km-E simulation, we start with the analysis of the large-scale circulation characteristics. The WAM winds are predominantly driven by the near-surface

pressure gradient between the heat low over the warm African continent and the high pressure system over the colder Gulf of Guinea (Thorncroft et al., 2011; Nicholson, 2013).

The pressure gradient between the Sahara heat low (SHL) and the high pressure system over the Gulf of Guinea is stronger in the 5 km-E simulation. This can be seen in Figure 5, which shows the mean 925 hPa geopotential height and the mean wind field at 925 hPa for the 40 km-P and 5 km-E simulation, respectively. The stronger high pressure system over the tropical At-

lantic in the 5 km-E simulation compared to the 40 km-P simulation, leads to a stronger pressure gradient in the Gulf of Guinea and to stronger winds in the Gulf of Guinea (Fig. 5 c). These winds in the Gulf of Guinea modulate the moisture transport into central Africa and support a stronger monsoon in the 5 km-E simulation, which cannot directly explain our previous findings of a weaker monsoon propagation in the 5 km-E simulation.

In contrast, the pressure gradient between the subtropical east Atlantic and the SHL are stronger in the 40 km-P simulation.

In the latter simulation the SHL and the high pressure system in the subtropical east Atlantic are stronger compared to the 5 km-E simulation. In addition to the stronger pressure systems, the SHL extends further west in the 40 km-P than in 5 km-E simulation, leading to an even stronger pressure gradient along the coast of Morocco and in the subtropical east Atlantic. The stronger pressure gradient accelerates the wind along the coast of Morocco and West Sahara in the 40 km-P simulation (Fig. 5 c) and results in a stronger wind convergence in the tropical Atlantic along 10 ° N. This wind convergence is important for

the moisture transport into the west Sahara. The stronger winds along 10 ° N indicate a stronger moisture transport into this region in the 40 km-P than in the 5 km-E simulation.

The northward propagation of precipitation during the West African Monsoon depends not only on the strength of the south-westerly low-level monsoon flow, but also on the strength of the Harmattan and the vertical lifting of airmasses throughout the troposphere. The northerlies associated with the hot and dry Harmattan winds are stronger in the 5 km-E than in the 40 km-

P simulation and counteract more strongly against the southerly monsoon flow. This surface convergence zone between the southerly monsoon flow and the northerly Harmattan is known as the Inner Tropical Front (ITF). In the 5 km-E simulation the ITF is located further south at around 17 ° N to 18 ° N, while in the 40 km-P simulation it is located at around 20 ° N (Fig. 6). The more northerly location of the ITF, as well as the generally weaker northward component against and above the surface monsoon winds, support the development of convection and therefore the higher precipitation rates further north in the 40 km-P

than in the 5 km-E simulation (Fig. 4 a).

In the JAS-mean the vertical wind is stronger in the 40 km-P simulation compared to the 5 km-E simulation (Fig. 6). Additionally, the ascent region is broader in the 40 km-P simulation compared to the 5 km-E simulation. This, on average, stronger and broader ascent region is consistent with stronger precipitation in the 40 km-P simulation as opposed to the 5 km-E simulation.

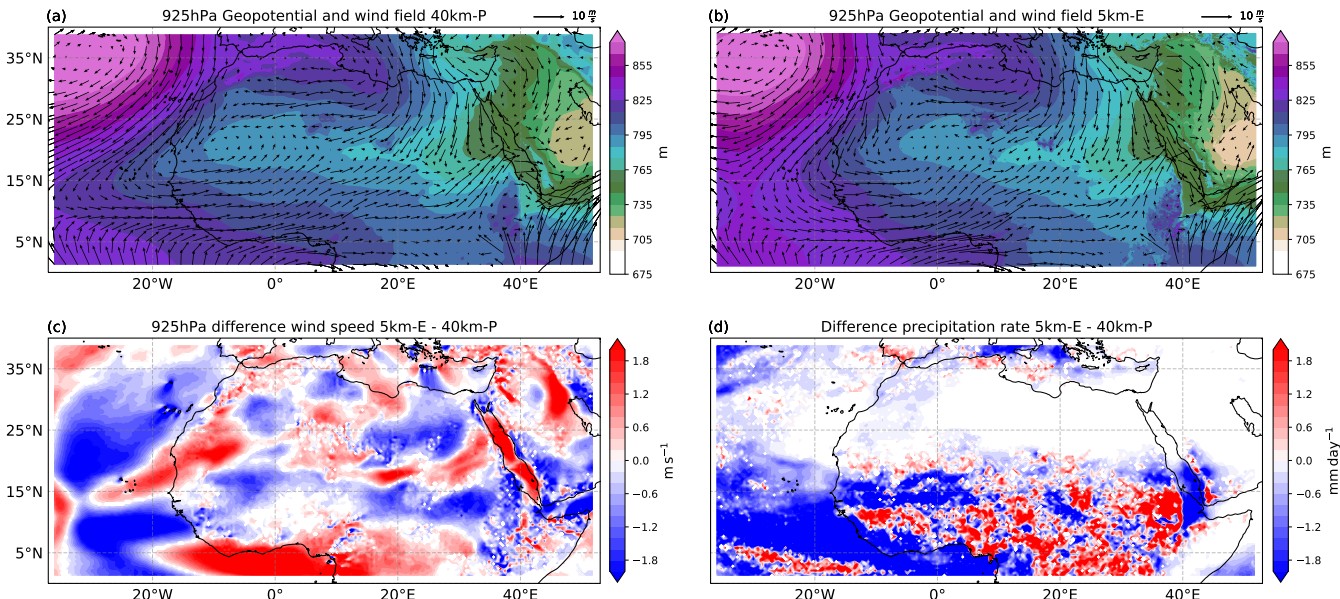

**Figure 5.** JAS mean geopotential height (shading) and mean wind field (vectors; m s$^{-1}$) at 925 hPa for the 40 km-P (a) and for the 5 km-E simulation (b), the difference in JAS mean wind speed at 925 hPa (c) and the difference in JAS mean precipitation rate (d) between the 5 km-E and the 40 km-P simulation, respectively. White colors in panel (c) and (d) display difference values between -0.1 and 0.1.

The strongest vertical velocities are located between 5° N and 15° N in the 40 km-P simulation and between 8° N and 15° N in the 5 km-E simulation. They are associated with the lifting between the Tropical Easterly Jet at around 5° N and at 200 hPa height and the African Easterly Jet (AFJ) at around 16° N and at 600 hPa height. A second weaker updraft region is located between 17° N and 20° N and is associated with the lifting of air masses at the ITF. We want to comment on the fact that we find a slightly weaker AEJ in the 5 km-E simulation compared to the 40 km-P simulation. E.g. Cook (1999) and Nicholson and Grist (2001) associated a weaker AEJ with wetter conditions over Africa in present-day conditions, which is contradictory to our results. Grist and Nicholson (2001); Nicholson and Grist (2001) suggest that the location of the AEJ is more important than its intensity. They link a more northern location of the AEJ core with higher rainfall and rainfall further north. Fig. 6 shows that the AEJ core in our simulations is located slightly further north in the 40 km-P than in the 5 km-E simulation. This is consistent with the higher precipitation rates and the slightly stronger northward extention of monsoonal rainfall in the 40 km-P compared to the 5 km-E and with the findings from Grist and Nicholson (2001); Nicholson and Grist (2001).

In conclusion, the stronger horizontal monsoon circulation (southwesterlies), the more northward location of the ITF, as well as the stronger and broader ascent region in the 40 km-P simulation as compared to the 5 km-E simulation, are all consistent with a stronger monsoon and a more northward propagation, in agreement with Fig. 4. Only the pressure gradient between the Gulf of Guinea and the SHL is stronger in the 5 km-E simulation. As will be shown later, the stronger pressure gradient

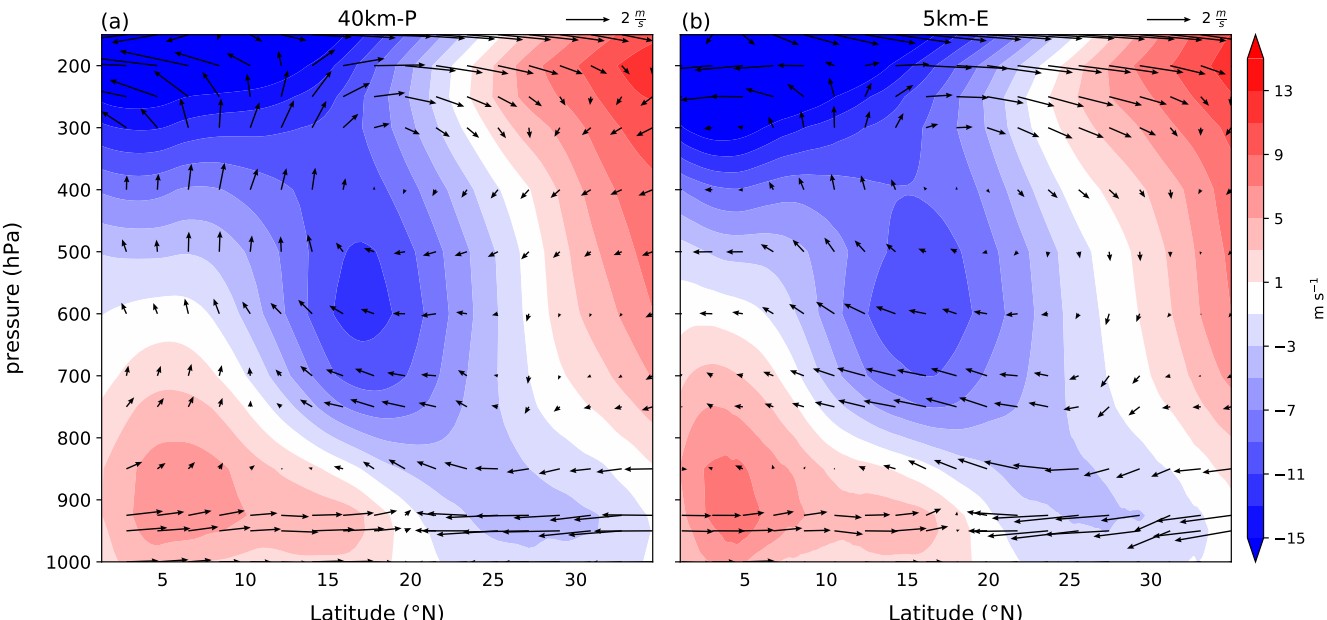

**Figure 6.** JAS mean crosssection of the zonally averaged wind field for the 40 km-P (a) and for the 5 km-E simulation (b). The shading shows the zonal wind in $m\,s^{-1}$. The vectors show the meridional and vertical wind field ($m\,s^{-1}$), where the vertical wind component is multiplied with 100 to make the vectors better visible. The mean is taken over all points of the dashed "WA-domain" outlined in Fig. 1.

between the Gulf of Guinea and the SHL does transport more moisture onto the African continent in the 5 km-E simulation, an effect nevertheless overcompensated by an excessively strong local drying of the African continent due to high amounts of runoff (see Sect. 3.4 and Sect. 3.5).

### 3.3 Thermodynamics

The large-scale monsoon circulation supports the higher precipitation rates in the 40 km-P simulation than in the 5 km-E simulation. However, if and how the prevailing atmospheric conditions lead to the development of convection and precipitation are essentially determined by the thermodynamic structure of the atmosphere. To investigate this, we examine the thermodynamical profiles for both representations of convection. We look at thermodynamical profiles for three different regions of north Africa as outlined in Fig. 1: b) the "coastal" region, c) the "sahel" region and d) the "sahara" region. Fig. 7 shows the corresponding thermodynamical profiles for 8 th September 7023 BP at 12 UTC. We choose 8th September as beeing representative of the prevalent state of the atmosphere for both representations of convection during JAS, as confirmed further below with Tabel A1 .

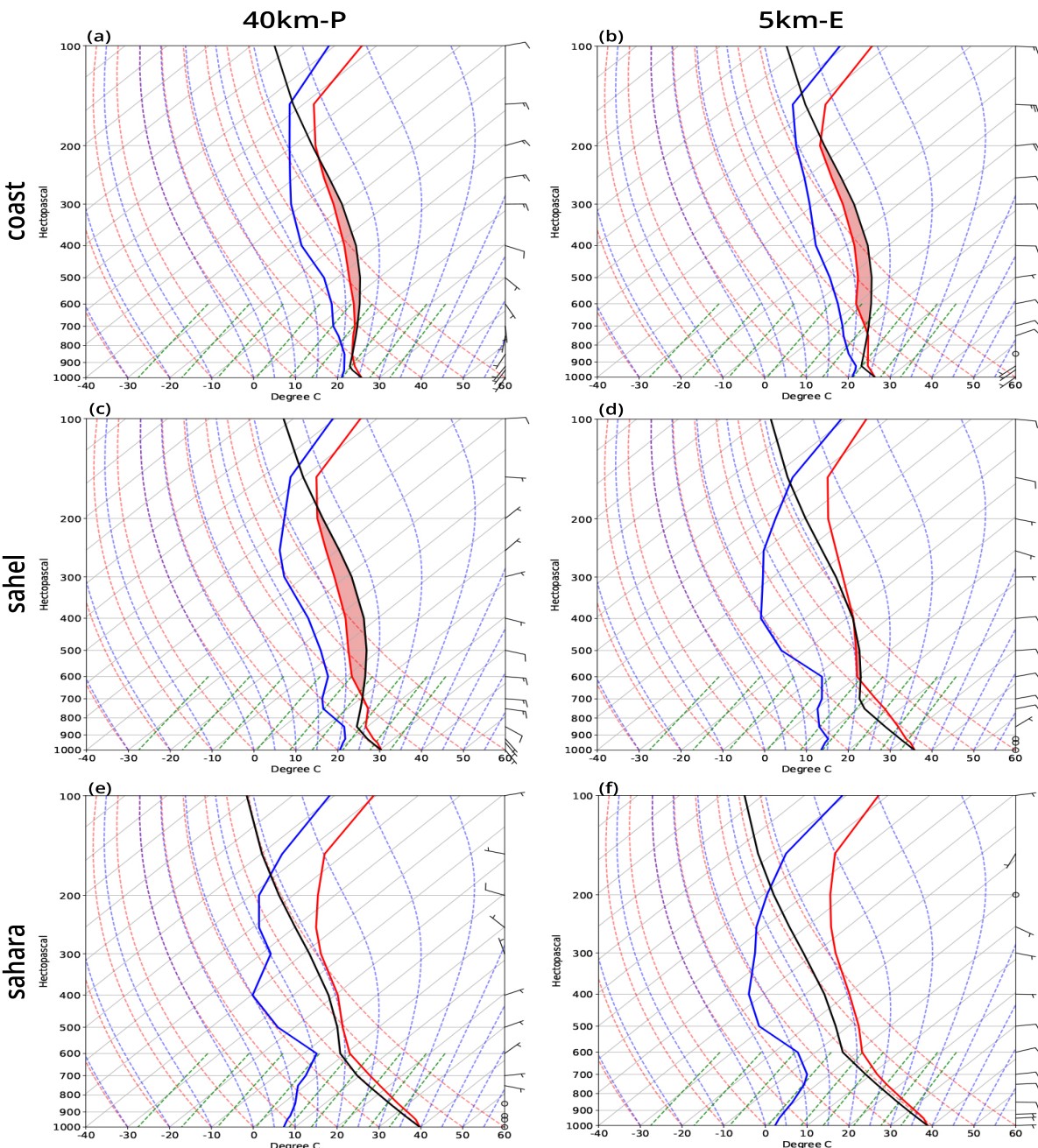

**Figure 7.** Skew-T Diagramms for the 8th September 7023 BP 12 UTC for the 40 km-P (left column) and the 5 km-E (right column) simulation. The red line depicts the temperature profile, the blue line the dew point and the black line the shows the path an air parcle would take through the atmosphere. The profiles are averaged over the three domains outlined in Fig. 1 labeled with "coast", "sahel" and "sahara". The red shaded area displays the CAPE. The lines in the background refer to the dry adiabats (red dashed), moist adiabats (blue dashed), isotherms (solid grey tilted) and isobars (solid grey horizontal).

|  |  | 40 km-P | 5 km-E |
|---|---|---|---|
| coastal | CAPE (J kg$^{-1}$) | 877.9 | 869.6 |
| | CIN (J kg$^{-1}$) | -27.1 | -43.6 |
| | T - T$_d$ (°C) | 5.1 | 6.1 |
| sahel | CAPE (J kg$^{-1}$) | 868.8 | 431.3 |
| | CIN (J kg$^{-1}$) | -172.0 | -221.5 |
| | T - T$_d$ (°C) | 13.4 | 17.3 |
| sahara | CAPE (J kg$^{-1}$) | 46.8 | 11.9 |
| | CIN (J kg$^{-1}$) | -120.8 | -65.8 |
| | T - T$_d$ (°C) | 30.6 | 32.8 |

**Table 1.** JAS-mean values of CAPE, CIN and dew point depression for the coastal, sahel and sahara region (Fig. 1).

The thermodyamic profiles over the coastal region show a higher level of convective inhibition (CIN) in the 5 km-E simulation compared to the 40 km-P simulation. As both, the 40 km-P and the 5 km-E simulation have a similar mean surface
temperatures around 26 ° C and mean dew point temperatures of around 21 ° C, the higher CIN in the 5 km-E simulation is a result of a more stably stratified atmosphere between 900 and 700 hPa. Combined with the weaker prevailing vertical velocity in the 5 km-E simulation (see Sect. 3.2), we conclude that convection can be less easily triggered in the 5 km-E simulation. The atmospheric conditions get even less unsupportive for convection in the 5 km-E simulation, when approaching the Sahara region. Over the Sahel, the 5 km-E simulation becomes even drier, consistent with a much higher surface temperature and low
dew point. The combination of a warm temperature profile and a low dew point temperature raises the Lifting Condensation Level (LCL) and the Level of Free Convection (LFC). Therefore, even if convection is triggered in the 5 km-E simulation, clouds will not deepen past 400 hPa height. From this, we would expect only little precipitation. The 40 km-P simulation stays moister in the Sahel region compared to the 5 km-E simulation. Moreover, the LCL in the 40 km-P simulation is lower and the convection, if triggered, can become very deep. The resulting precipitation is likely to be stronger than in the 5 km-E simula-
tion.
Over the Sahara, both the 40 km-P and the 5 km-E simulation become even drier and surface temperatures rise. The lack of moisture and the stably stratified atmosphere in both simulations reveal that it becomes very unlikely that convection is triggered and precipitation develops in this region.
Table 1 lists the mean values of CAPE, CIN and the dew-point depression for the three regions to emphasize that the results
are valid over the whole JAS season. These findings emphasize that the moisture availability and the thermodynamical state of the atmosphere is much more supportive for the development of convection in the 40 km-P simulation than in the 5 km-E simulation. Together with the stronger vertical motion in the 40 km-P simulation (see Sect. 3.2), this is consistent with the higher

precipitation rates in the 40 km-P simulation. Furthermore, this suggests that, beside the large-scale circulation and the stability of the atmosphere, the availability of moisture in the two simulations also contributes to the differences in precipitation. We

turn our attention to the availability of moisture in the 40 km-P and the 5 km-E simulation in the next two sections.

### 3.4 Moisture Field and Moisture Transport

The thermodynamical profiles revealed more humid conditions in the 40 km-P simulation compared to the 5 km-E simulation, especially in the semi-arid transition zone of the Sahel region. Now, we investigate the moisture field in more detail. The moisture field supports the findings from the previous section that the 40 km-P simulation is overall moister over the continent

than the 5 km-E simulation (Fig. 8). The vertical cross section of specific humidity in Fig. 8a and b shows higher amounts of water vapour in the planetary boundary layer in the 40 km-P simulation compared to the 5 km-E simulation, especially between $15\,°\,N$ and $25\,°\,N$. This region coincides with the region where we found higher precipitation rates in the 40 km-P than in the

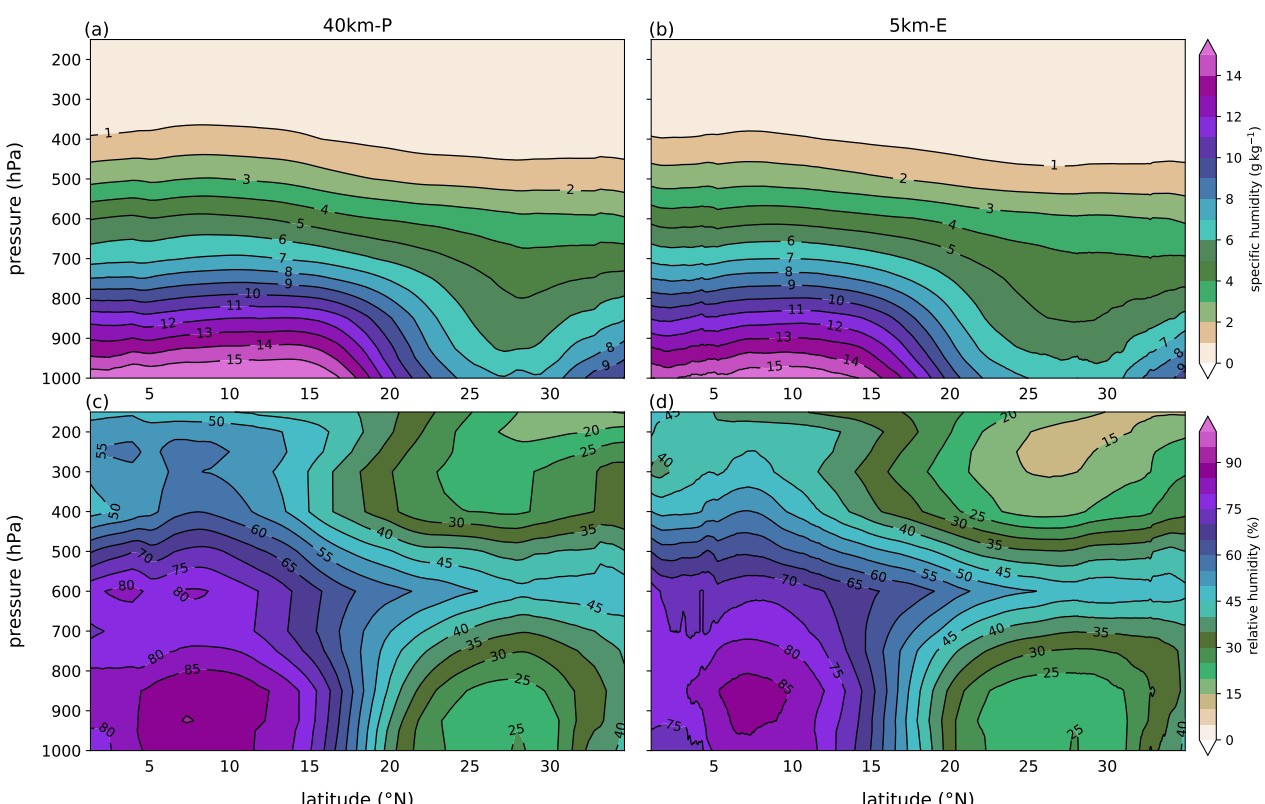

**Figure 8.** JAS mean vertical crosssection of specific humidity in $g\,kg^{-1}$ in the two upper panels (a,b) and for relative humidity in % in the two bottom panels (c,d) for the 40 km-P (a,c) and the 5 km-E (b,d) simulation, respectively. The mean is taken over land of the dashed domain outlined in Fig. 1 a.

The vertical cross section of relative humidity displays a deep core of moist air at around 8 ° N in both simulations, extending
from the surface into the upper troposphere. This region coincides with the region of the strongest vertical velocities (Fig. 6).
Throughout the troposphere the moisture in the deep core exceeds 50 %, whereas over the Sahara desert the relative humidity
is much lower. Comparing the two simulations, the 40 km-P simulation shows higher values of relative humidity throughout
the depth of the troposphere. This would tend to favour precipitation in the 40 km-P simulation.

There are two possible mechanisms for supplying moisture for precipitation over the continent: 1) advection of moisture from
surrounding regions and 2) local evapo(transpi)ration (see Sect. 3.5). This provides two possible explanations for the on-average
wetter atmosphere in the 40 km-P simulation. Either the north African continent receives more moisture through the moisture
transport from the ocean and/or moisture recycling over land is more effective in the 40 km-P than in the 5 km-E simulation.

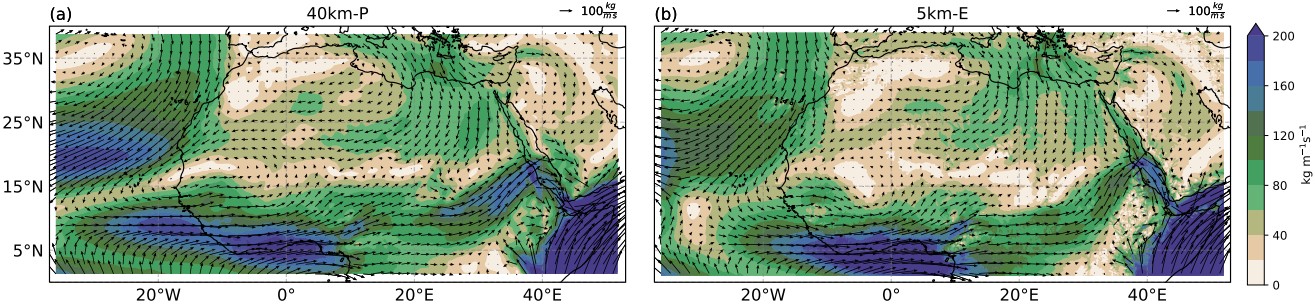

**Figure 9.** JAS 1000 hPa to 850 hPa vertically integrated moisture flux magnitude (shading) and mean vertically integrated moisture flux
(vectors; kg m$^{-1}$ s$^{-1}$) for the 40 km-P (a) and the 5 km-E (b) simulation.

First we look at the moisture transport. Figure 9 shows the JAS vertically integrated moisture flux magnitude and moisture
flux for the 40 km-P and the 5 km-E simulation. We integrate the lower atmosphere levels from 1000 hPa to 850 hPa. The
stronger winds in the tropical Atlantic in the 40 km-P simulation (see Fig. 5 c and section 3.2), which are associated with the
African Westerly Jet, result in a stronger moisture transport from this region into the west Sahel-Saharan region compared
to the 5 km-E simulation. Furthermore, the moisture transport originating from the Mediterranean sea towards the Sahara is
stronger in the 40 km-P simulation compared to the 5 km-E simulation. However, and more importantly, the moisture transport
from the Gulf of Guinea, which supplies moisture dominantly into central north Africa, is stronger in the 5 km-E simulation
compared to the 40 km-P simulation due to the stronger winds in this region (compare to Fig. 5 c and Sect. 3.2). The result
that the tropical Atlantic along 10 ° N and the Gulf of Guinea supply moisture for the west Sahel-Saharan region and central-
north Africa, respectively, is consistent with the results from Druyan and Koster (1989) and Lélé et al. (2015) for present-day
conditions. Over the WA-Domain the domain-mean moisture flux is 84.2 kg m$^{-1}$ s$^{-1}$ and 81.8 kg m$^{-1}$ s$^{-1}$ for the 40 km-P
and the 5 km-E simulation, respectively. Hence, in a mean sense, there is a slightly larger domain-mean moisture flux over the
WA-Domain in the 40 km-P simulation compared to the 5 km-E simulation.

## 3.5 Land-atmoshpere coupling

Besides the moisture transport from the ocean and humid coastal regions into north Africa, the local source of moisture due to evapotranspiration needs to be considered as well. The evapotranspiration (Fig. 10 a) coinincides with the precipitation rates (Fig. 4 a) in the 40 km-P and the 5 km-E simulation. In other words, between the equator and 8 ° N the evapotransipration is higher in the 40 km-P simulation than in the 5 km-E simulation.

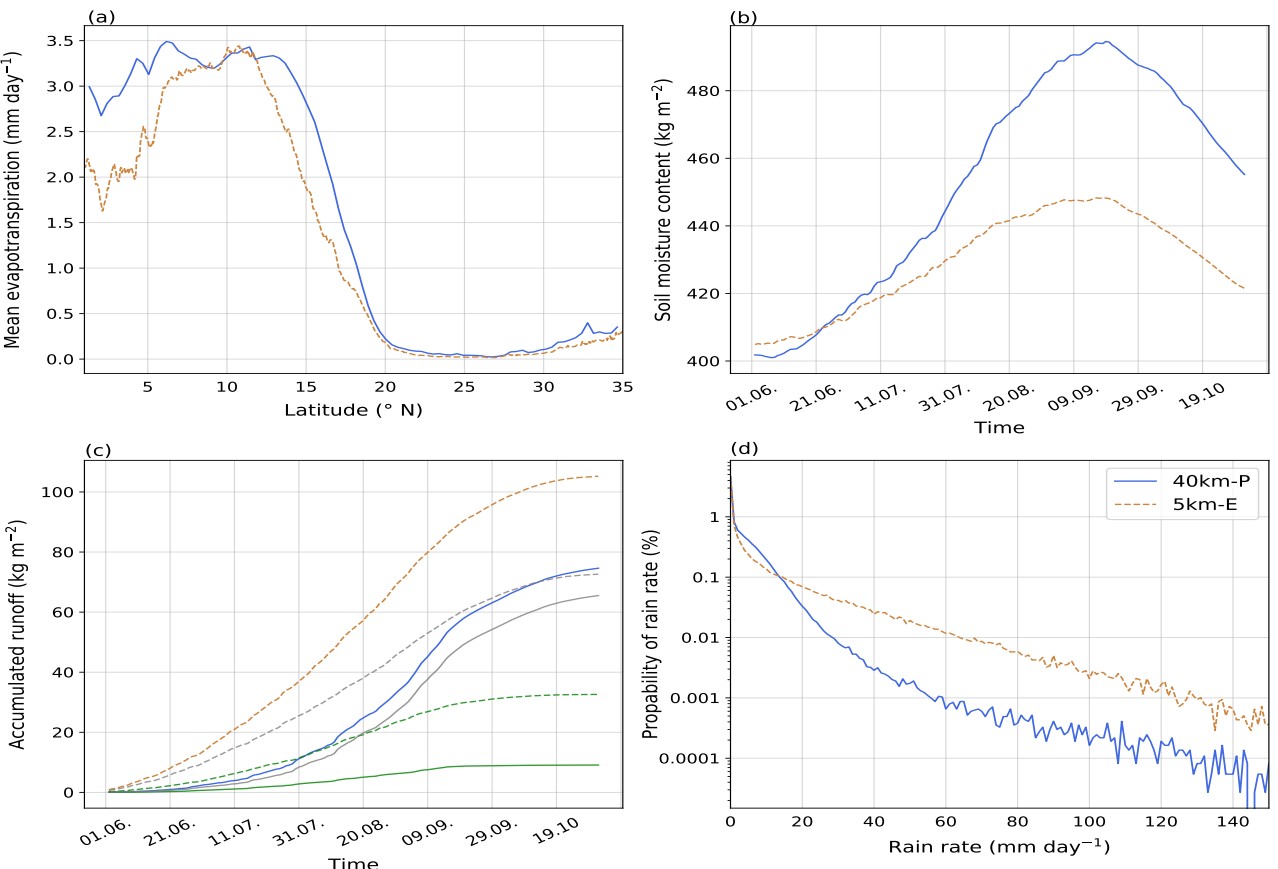

**Figure 10.** a) JAS mean meridional distribution of evaporation b) timeseries of soil water content: sum over the first 6 layers (1.62 m), c) JJASO timeseries of accumulated runoff, surface runoff (green lines) and groundwater runoff (grey lines), d) Propability Density Function (PDF) of JAS mean rain rate for the 40 km-P (blue-solid) and the 5 km-E (orange-dashed) simulation excluding days with rain rates equal to 0 mm day$^{-1}$. All calculations are performed over land of the dashed "WA-domain" outlined in Fig. 1.

Between 8 ° N and 12 ° N, evapotranspiration is equally strong in both simulations, and north of 12 ° N it becomes larger in the 40 km-P simulation again. The regions of higher evapotranspiration in the 40 km-P simulation reflect the higher precipitation rates in the 40 km-P simulation compared to the 5 km-E simulation.

The evaporation is strongly coupled to soil moisture, especially in regions with sparse or no vegetation. The soil moisture up to 1.62 m depth is much lower in the 5 km-E simulation compared to the 40 km-P simulation especially during our analysis period from July to September (Fig. 10 b). The lower soil moisture in the 5 km-E simulation is due to higher amounts of total runoff (Fig. 10 c) compared to the 40 km-P simulation. The higher runoff is due to both, higher surface and higher groundwater runoff, where groundwater runoff refers to the sub-surface runoff in all soil layers. The ratio between surface to groundwater

runoff is higher in the 5 km-E compared to the 40 km-P simulation, as it is expected from the precipitation characteristics (see next paragraph). The high amount of runoff in the 5 km-E simulation prevents the remoistening of the soil by precipitation, maintaining low levels of soil moisture and leading to a overly dry and warm atmosphere for convection to develop efficiently. Based on results from Savenije (1996), the runoff plays a key role for the recycling of water over semi arid regions (i.e the Sahel-Saharan region).

The differences in the amount of runoff in the 40 km-P and the 5 km-E simulation are driven by substantial differences in the characteristics of the precipitation distribution. Figure 10 d displays the propability that a simulation produces dominantly light (low rain rates) or stronger precipitation events (high rain rates). The figure reveals that in the parameterized simulations it is much more likely to produce light rainfall ($< 15 \, \mathrm{mm \, day}^{-1}$), while the simulation with explicitly resolved convection exhibits much more intense precipitation events ($> 15 \, \mathrm{mm \, day}^{-1}$). The light rainfall in the 40 km-P simulation is widespread over the

whole domain, while the precipitation in the 5 km-E simulation occurs more locally (not shown). High amounts of precipitation in a short time interval and over a small area in the 5 km-E simulation lead to the high amounts of runoff, as the water uptake by the soil is limited. On the contrary, in the 40 km-P simulation the constantly light and large-scale precipitation sufficently moistens the soil throughout the simulation period.

## 3.6    Diurnal cycle

Marsham et al. (2013) identified the difference in the timing of the precipitation diurnal cycle between parameterized and explicit convection as the main driver for the differences in the meridional distribution of precipitation in their present-day simulations. We test whether this effect has an impact in our simulations. We find that the precipitation in the 40 km-P modified ("MOD") setup is neither weaker nor shifted further southward compared to the 5 km-E and the 40 km-P simulation (Fig. 4 b). Moreover, the 40 km-P$_{MOD}$ simulation exhibits even more precipitation, especially between the equator and 15 ° N

compared to the other simulations. We conclude that a later precipitation peak does not favor a more northward propagation of precipitation in our model framework as was the case in Marsham et al. (2013).

The differences between our results and the results of Marsham et al. (2013) can be due to various aspects related to the model and simulation setup. Firstly, the model Marsham et al. (2013) used, utilizes a different convective parameterization than ICON-NWP. ICON-NWP uses the same convection scheme as the IFS, a convection scheme that has been continuously

developed and tuned to best match today's precipitation distribution.

Secondly, the analyzed period in our study is three months, while Marsham et al. (2013) focused on a 10-day period. They chose their simulation period during the peak of the monsoon season from the end of July until the beginning of August. We also find 15-day periods in our simulations where the 5 km-E simulation propagates further north than the 40 km-P simulation.

This suggests that the northward extent of monsoonal precipitation is very variable on short timescales.

Thirdly, we simulate a much larger domain, covering the whole north African continent and parts of the Atlantic ocean, while Marsham et al. (2013) focused on a smaller land-only domain from $10\,^\circ$ E to $10\,^\circ$ W and $5\,^\circ$ N to $25\,^\circ$ N. The latter two facts imply that different characteristics and amounts of precipitation from different regions in Africa, as well as the large-scale circulation and effects from the Atlantic ocean influence our analysis. These effects are are not captured in the study of Marsham et al. (2013). Berthou et al. (2019) performed a 10-year study comparing simulations with explicit and parameterized

convection performed with the Met Office Unified Model. This is the same model which was used in Marsham et al. (2013). In the 10-year mean, the simulations with explicitly resolved convection did not show a substantially stronger northward extent of precipitation than the parameterized ones, a result closer to our findings.

## 4   Summary and Conclusion

In this study, we investigated whether the representation of convection (parametrized versus explicit) impacts the meridional

distribution of monsoonal rainfall under mid-Holocene atmospheric conditions (i.e. orbital parameters, tracer gases) over north Africa. For that purpose we ran regional, nested simulations with the atmospheric model, ICON-NWP. To analyse the meridional distribution of precipitation in both settings, we compared 40 km parameterized (40 km-P) with 5 km explicitly resolved convection (5 km-E) simulations. Furthermore, we isolated the impact of different resolutions from those of different representations of convection by comparing 10 km parameterized (10 km-P) and explicitly resolved convection (10 km-E) simulations.

In agreement with the results of previous studies conducted for present-day conditions (Marsham et al. (2013) , Dirmeyer et al. (2012), Pearson et al. (2014)), the precipitation distribution across simulations with the same representation of convection are more similar than to simulations with the same grid spacing.

Marsham et al. (2013) found a stronger northward propagation of precipitation in explicit convection simulations compared to parameterized simulation for present-day conditions. This motivated our study and raised the question: Does the represen-

tation of convection also impact the northward extent of the West African Monsoon (WAM) during the mid-Holocene? In the JAS-mean, our 40 km-P simulation produces around $0.8\,\mathrm{mm\,day^{-1}}$ per latitude more precipitation north of $12\,^\circ$ N than the 5 km-E simulation. As such, the representation of convection does impact the northward extent of the WAM, but in the opposite way initially thought, with a stronger propagation in the parameterized simulation. Compared to the results of Marsham et al. (2013) this is mainly because the parametrization of convection in ICON-NWP produces already a more realistic meridional

distribution of precipitation than the Met Office Unified Model.

The differences in the meridional precipitation distribution between explicitly resolved convection and parameterized convection in our simulations is caused by thee factors:

– We identified a generally stronger monsoonal circulation over the north African continent in the 40 km-P than in the 5 km-E simulation. The near surface southwesterly monsoon flow over land is stronger in the 40 km-P than in the 5 km-

E simulation. Furthermore, in the 5 km-E simulation the northerlies from the hot and dry Harmattan counteract more strongly the monsoonal winds. These northerlies push the Inner Tropical Front (ITF) southward. We also found that the

(positive) vertical component of the wind field is, in the JAS-mean, stronger overall the analyzed domain in the 40 km-P than in the 5 km-E simulation.

– The thermodynamic structure of the atmosphere in the 40 km-P simulation is more supportive for the development of clouds and precipitation. The convective inhibition is lower in the 40 km-P simulation compared to the 5 km-E simulation, due to the atmosphere being less stable.

– The 40 km-P simulation is moister than the 5 km-E simulation. This is especially true for the region between 15 ° N and 25 ° N, which coincides with the region of higher precipitation in the 40 km-P simulation compared to the 5 km-E simulation. The strength of moisture transport from the ocean to the African continent depends on the ocean region; over the Gulf of Guinea the moisture transport is stronger in the 5 km-E simulation, but the moisture transport from the tropical east Atlantic is stronger in the 40 km-P simulation. More importantly, we found more evapotranspiration in the 40 km-P than in the 5 km-E simulation. The higher evapotransipration rate is due to a higher soil moisture content throughout the whole simulation period. This is due to much weaker runoff in the 40 km-P than in the 5 km-E simulation. These differences in runoff result from substantially different precipitation characteristics. In the 40 km-P simulation, light precipitation can be stored by the soil more easily. The moister soils favor evapotranspiration which then makes it easier to trigger convection and to produce precipitation. In contrast, the 5 km-E simulation exhibits much more intense precipitation events which occur less frequently and more locally, producing strong runoff and preventing the soil moisture from being refilled by precipitation. The drier conditions, especially in the transition zone of the Sahel region, hampers the development of convection and precipitation in the 5 km-E simulation compared to the 40 km-P simulation.

We conclude that using regional climate simulations using resolved, i.e. explicitly resolved, deep convection do not necessarily produce more precipitation in the mid-Holocene Sahara-Sahel region than simulations with parameterized deep convection. However, we have shown that the precipitation characteristics, in particular the absence of permanent light rainfall and more intense convective events in the simulations with resolved deep convection are closer to what one would expect.

Our study pinpoints to the key role that soil hydrology may take in controlling the amount of rainfall in simulations with explicitly resolved convection. On the one hand, the used land-surface scheme might be limited to store large moisture amounts in the soil. On the other hand, it is not able to create overground lakes and wetlands in mid-Holocene climate simulations. However, these local moisture sources in the mid-Holocene Sahara-Sahel region might involve important local feedback mechanisms in simulations with explicitly resolved convection. To investigate these possible limitations of our simulations, the atmosphere-soil hydrology interaction will be subject to further numerical experiments in which we will also include the effect of a more vegetated 'green Sahara' on the difference between simulations with resolved and parameterized deep convection.

*Code availability.* http://hdl.handle.net/21.11116/0000-0007-6597-D

## Appendix A: Analysis of 10 km simulations

Here, we present the figures, we analyzed in the main paper, again for the parameterized and explicitly resolved convection simulation with 10 km horizontal resolution.

In Fig. 4 (a) and in the corresponding section we argued that the meridional distribution of precipitation in the WA-Domain (Fig. 1) is more similar between the 40 km-P and 10 km-P simulation and between the 5 km-E and 10 km-E simulation, respectively, than between the 10 km-P and 10 km-E simlation. Here, we want to analyze the 10 km-simulations a bit more carefully. The 10 km-P simulation produces more precipitation than the 10 km-E simulation. Especially interesting for us is the higher precipitation north of around 13 °N. Similar to the 40 km-P simulation, the 10 km-P simulation exhibits a double precipitation peak, which is absent in the 10 km-E simulation. In the following, we show that the mechanisms described in the main paper also hold for the 10 km-P and 10 km-E simulation.

### A1 Large-scale circulation

Fig. A1 shows that the SHL is stronger and extents further west in the 10 km-P than in the 10 km-E simulation. Different from the 40 km-P and 5 km-E, the pressure gradient between the SHL and the high pressure system over the Gulf of Guinea is stronger in the 10 km-P than in the 10 km-E simulation (Fig. A1 a and b). This drives stronger winds in the Gulf of Guinea in the 10 km-P simulation.

In contrast, the pressure gradient between the SHL and the high pressure system in the east subtropical Atlantic is slightly stronger in the 10 km-E simulation, accelerating the wind along the West African coast and int the Atlantic between 10 °N to around 24 °N. However, the African westerly jet (at around 5 °N to 10 °N) is stronger in the 10 km-P compared to the 10 km-E simulation, similar to the results in the 40 km-P and 5 km-E simulation. The differences in the wind speed (Fig. A1 c) and in precipitation (Fig. A1 d) are very similar to those in (Fig. 5 c and d). Fig. A2 shows that the low-level monsoon winds are equally strong or only slightly stronger in the 10 km-P compared to the 10 km-E simulation. The dry Harmattan, antagonizing the monsoon flow, is slightly stronger in the 10 km-E pushing the ITF further south (to around 17 °N) in the 10 km-E simulation compared to the 10 km-P simulation (ITF at around 20 °N).

Strongly similar to Fig. 6 is the difference in the strength of the vertical ascent between the 10 km-simulations. In the 10 km-P simulation, the vertical upward wind is substantially stronger in the mean than in the 10 km-E simulation, consistently with the results we find in the 40 km-P and the 5 km-E simulation. The stronger upward motion and the further northward located ITF are supportive for convection and precipitation in the parameterized simulation.

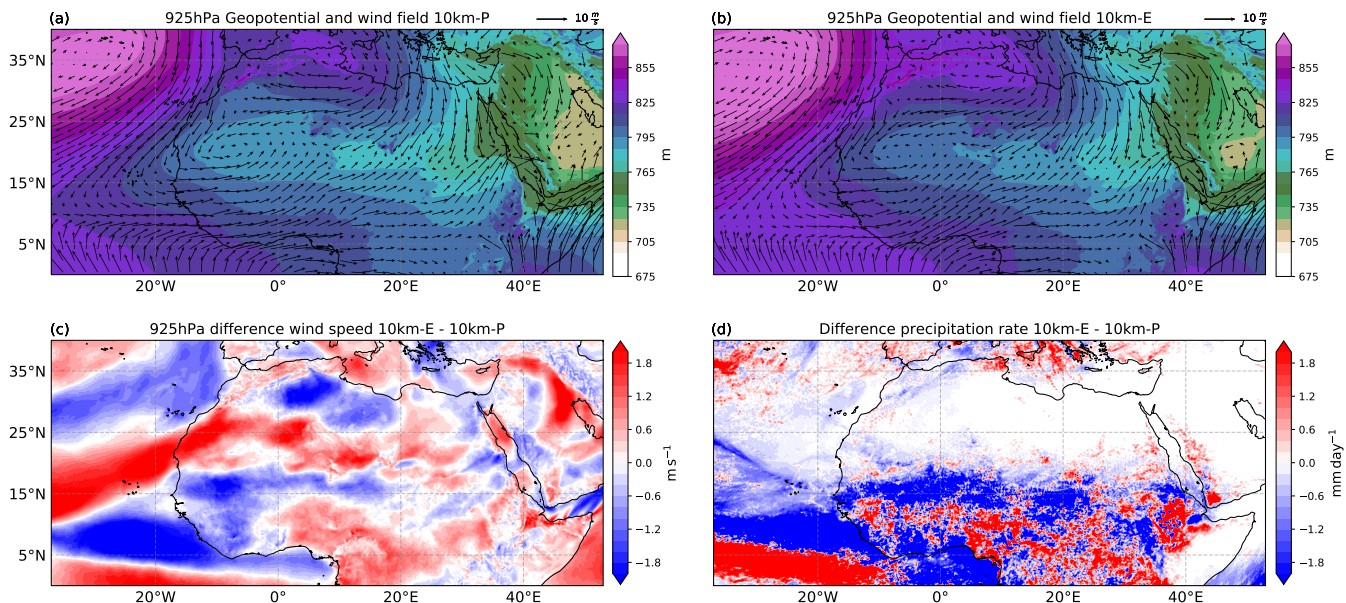

**Figure A1.** The same as Fig. 5 but for the 10km-P and 10 km-E simulation.

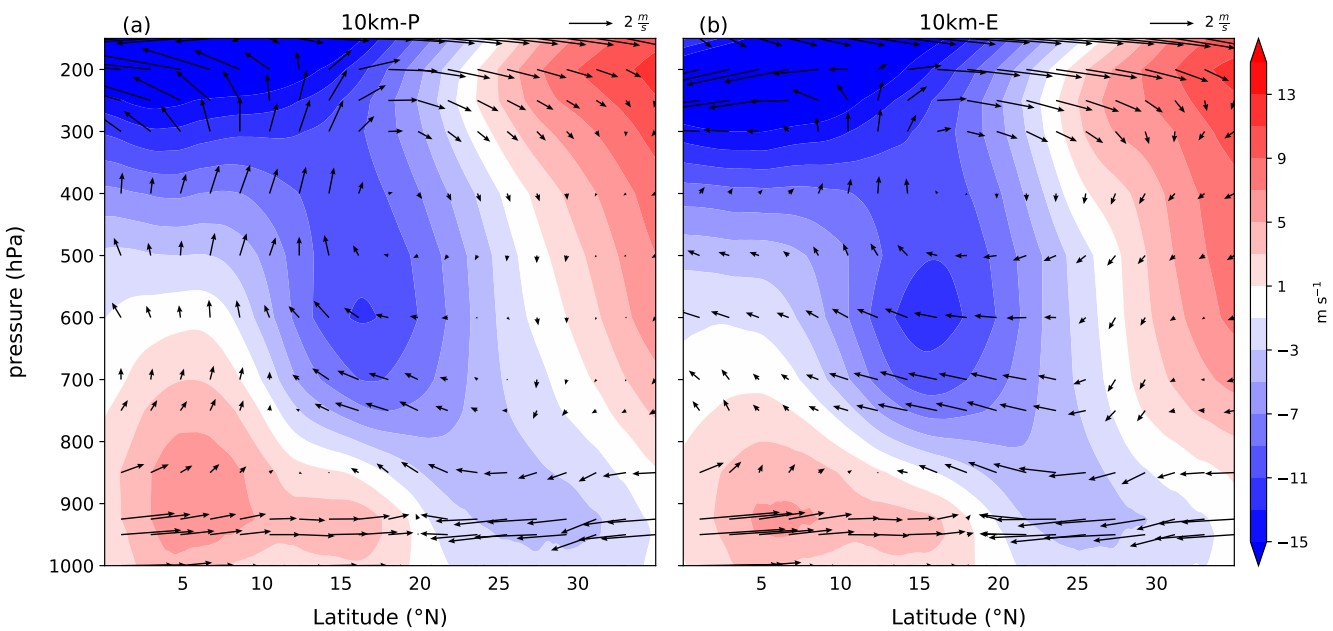

**Figure A2.** The same as Fig. 6 but for the 10km-P and 10 km-E simulation.

## A2 Thermodynamics

We present the mean CAPE and CIN values in Table A1 similar to Table 1 in Sec. 3.3. We show that the thermodynamic structure of the atmosphere in the 10 km-P and the 10 km-E simulation is similar as in the 40 km-P and 5 km-E simulation, respectively (Sec. 3.3). In particular the differences between explicit and parameterized convection over the Sahel, with a larger dewpoint depression and an atmosphere less conductive to convection is also clearly visible when comparing 10kmP to 10km-E.

|  |  | 10 km-P | 10 km-E |
|---|---|---|---|
| coastal | CAPE ($J\,kg^{-1}$) | 771.5 | 942.6 |
| | CIN ($J\,kg^{-1}$) | -29.7 | -35.9 |
| | $T - T_d$ (°C) | 5.3 | 5.8 |
| sahel | CAPE ($J\,kg^{-1}$) | 772.3 | 381.2 |
| | CIN ($J\,kg^{-1}$) | -187.6 | -227.5 |
| | $T - T_d$ (°C) | 13.6 | 17.6 |
| sahara | CAPE ($J\,kg^{-1}$) | 60.8 | 5.7 |
| | CIN ($J\,kg^{-1}$) | -121.9 | -31.4 |
| | $T - T_d$ (°C) | 29.9 | 32.9 |

**Table A1.** The same as table 1 but for the 10 km-P and 10 km-E simulation.

## A3 Moisture field and moisture tranport

The more humid conditions revealed by the thermodynamics are confirmed by the vertical moisture crosssection (Fig. A3) of specific (a-b) and relativ humidity (c-d). In the lowest atmosphere levels the 10 km-P simulation is more humid than the 10 km-E simulation, especially north of 15 °N - the transition region where we also find the higher precipitation rates (Fig. 4).

The horizontal transport of moisture from the Atlantic and the Gulf of Guinea region towards the African continent is higher in the 10 km-P than in the 10 km-E simulation, constistent with the moisture crosssection (Fig. A3). The moisture import from the mediterranean sea is similar in both simulations. The mean moisture transport in the WA-Domain is 84.5 $kg\,m^{-1}\,s^{-1}$ and 69.6 $kg\,m^{-1}\,s^{-1}$ in the 10 km-P and 10 km-E simulation, respectively. Therefore, the higher moisture transport supports the generally moister conditions revealed by Fig. A3.

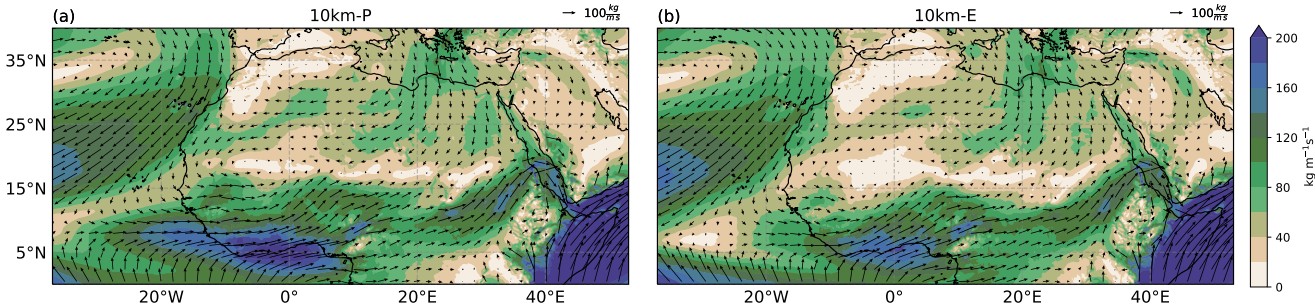

**Figure A3.** The same as Fig. 8 but for the 10km-P and 10 km-E simulation.

**Figure A4.** The same as Fig. 9 but for the 10km-P and 10 km-E simulation.

## A4 Land-Atmosphere Coupling

The coupling of the atmosphere and the surface are very similar as described in Sec. 3.5. Fig. A5 (a) shows higher evapotranspiration over land at all latitudes in the 10 km-P compared to the 10 km-E simulation. The higher evaporation is consistent with the higher soil moisture values shown in panel b. The higher evapotranspiration rates in the 10 km-P simulation are consistent with weaker surface and groundwater runoff compared to the 10 km-E simulation. With other words, the soil looses less water into runoff and stays moister. This is forced by differences in the precipitation intensity and its spatial distribution. Fig. A5 (d) shows that the 10 km-P simulation produces more often (higher propability values) light and widespread precipitation compared to the 10 km-E simulation.

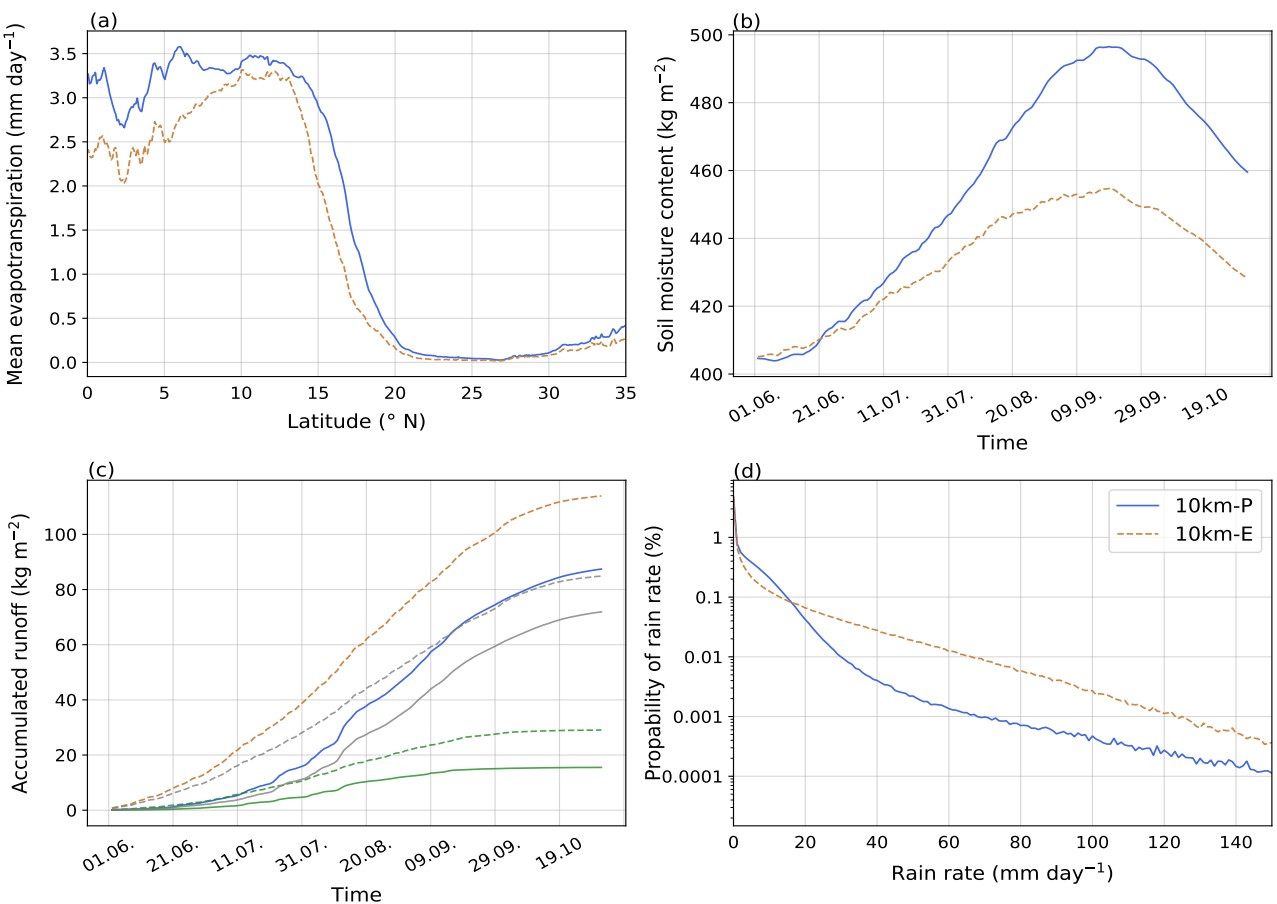

**Figure A5.** The same as Fig. 10 but for the 10km-P and 10 km-E simulation.

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

*Author contributions.*  CH and MC designed the research project and the experiments. LJ performed the simulations and analysis. CH, JB and MC gave input, ideas and feedback to the analysis of the simulations. LJ prepared the manuscribt with contributions from all co-authors.

*Competing interests.*  The authors declare that they have no conflict of interest.

*Acknowledgements.*  We thank Jürgen Bader and Anne Dallmeyer for valueable discussions and comments. Further we thank Roberta D'Agostino for internal review and acknowledge Reiner Schnur for technical support. This research was supported by the International Max Planck Research School on Earth System Modelling (IMPRS-ESM), Hamburg. The model simulations were performed at the Deutsche Klimarechenzentrum (DKRZ).