# Peer review of "Influence of the representation of convection on the mid-Holocene West African Monsoon"

_Climate of the Past, 2020_

## Referee Comment (RC2)

Review of "Influence of the representation of convection on the mid-Holocene West African Monsoon" by Leonore Jungandreas et al.

Reviewer: Aiko Voigt

Jungandreas et al. address the long-standing challenge of capturing the northward extension of the West-African monsoon during the mid-Holocene in climate models. The rainfall extension is indicated by climate proxies, but coarse resolution models with parametrized convection consistently have failed to capture it. One suspected reason is the misrepresentation of convection in such coarse models; i.e., it has been hypothesized that representing convection explicitly by going to storm-resolving resolutions might "solve" this problem. In this paper, the authors show that this is not the case, at least not in the ICON-NWP model in limited-area setup used here. Quite the contrary, they find that a low-resolution version of the model with parametrized convection exhibits a more northward precipitation extension than the fine-resolution version with explicit convection. This is an interesting and intriguing result, based on which I strongly support the publication of the paper in Climate of the Past. Another interesting finding is that the "failure" of the fine-resolution model version can be ascribed to the inability of the soils to hold the large amount of rainfall generated, leading to strong runoff, relatively drier soils, and hence less precipitation.

The paper is well written and clearly structured - this is much appreciated. A potential shortcoming of the paper is that some of the analysis could go into more detail, and it would seem they could do so with relatively little additional work. I give a few examples below. At the same time, I feel the results as they stand are sufficiently interesting, and so these examples are suggestions that the authors might or might not want to follow.

L8ff: I find the abstract to not be completely consistent. It starts with saying that the 5km-E version has a more realistic spatial distribution and intensity of precipitation, and then argues that the 40km-P version performs consistently better. I understand the point regarding the precipitation intensity, but not the point about the spatial distribution.

L65ff, Sect. 2.1: It would be nice to have a little more background on the simulation setup. E.g., what is the update frequency of the lateral boundary data?

L65ff, Sect. 2.1: I would also be interested in seeing how the ICON-NWP runs compare to the precipitation from the global MPI-ESM model. E.g., is 40km-P also better than MPI-ESM?

L94ff: I would like to see a bit more justification for the chosen years, especially since later only one of the years is studied in more detail. E.g., a figure would help to make the arguments more explicit.

L105: How is the diurnal cycle modified? And why is the 5km-E version also affected by this change (Fig. 2b)? It then seems the change cannot be a tuning parameter of the convection scheme.

L125: I find the wording of "per latitude" unnecessary or confusing. The units of precip are mm/day and not mm/day/latitude.

Fig. 3, caption: Domain a should probably read WAM domain.

L188: I assume the local drying refers to the runoff described later. Maybe this can be hinted at already here so as to help orient the reader?

Fig. 8: In addition to the maps it would be nice if you could calculate the moisture flux into/out of the WAM domain. It's a bit hard to see from the maps.

L249: I suggest you include a sentence of the end of this section that I assume should say that there is more moisture advection in the 5km run, so this cannot explain the drier atmosphere.

Data statement: I would like to see a proper data statement. Can the simulations be made public? I found the analysis scripts and the runs scripts in the linked data file. That should be described in more detail.
* * *
Typos:

L8: remove "," after precipitation

L9: I think the line break should be removed

L68: hasalready

L133: the the

L254: another->other

L327: The the

L342: Add "," after characteristics and after events (in line 343)

---

## Author Response (AR1)

Point-by-point response with all relevant changes made in the manuscript
cp-2020-162 "Influence of the representation of convection on the mid-Holocene West African Monsoon" by Jungandreas et al.

**Reviewer Aiko Voigt**

**Response to the review by Aiko Voigt on the manuscript cp-2020-162 "Influence of the representation of convection on the mid-Holocene West African Monsoon" by Leonore Jungandreas et al.**

We thank the reviewer for his effort in carefully reading and commenting on our manuscript. In the following, we reply to his comments point by point.

*Review: Jungandreas et al. address the long-standing challenge of capturing the northward extension of the West-African monsoon during the mid-Holocene in climate models. The rainfall extension is indicated by climate proxies, but coarse resolution models with parametrized convection consistently have failed to capture it. One suspected reason is the misrepresentation of convection in such coarse models; i.e., it has been hypothesized that representing convection explicitly by going to storm-resolving resolutions might "solve" this problem. In this paper, the authors show that this is not the case, at least not in the ICON-NWP model in limited-area setup used here. Quite the contrary, they find that a low-resolution version of the model with parametrized convection exhibits a more northward precipitation extension than the fine-resolution version with explicit convection. This is an interesting and intriguing result, based on which I strongly support the publication of the paper in Climate of the Past. Another interesting finding is that the "failure" of the fine-resolution model version can be ascribed to the inability of the soils to hold the large amount of rainfall generated, leading to strong runoff, relatively drier soils, and hence less precipitation.*

*The paper is well written and clearly structured - this is much appreciated. A potential shortcoming of the paper is that some of the analysis could go into more detail, and it would seem they could do so with relatively little additional work. I give a few examples below. At the same time, I feel the results as they stand are sufficiently interesting, and so these examples are suggestions that the authors might or might not want to follow.*

*L8ff: I find the abstract to not be completely consistent. It starts with saying that the 5km-E version has a more realistic spatial distribution and intensity of precipitation, and then argues that the 40km-P version performs consistently better. I understand the point regarding the precipitation intensity, but not the point about the spatial distribution.*

**Reply:** With the more realistic spatial distribution in the 5km-E simulations we refer to the occurrence of more local (but strong) precipitation events. In the 40km-P simulation we notice that there is almost no grid cell that receive any precipitation. This gives a spatial precipitation pattern of widespread (at least) light precipitation. In the 5km-E simulation it occurs much more often that grid cells receive no rain. This gives a spatial pattern that is more locally confined and not so extended as in the 40km-P simulation. We assume the fact that it is not always and everywhere drizzling in the 5km-E compared to the 40km-P simulation to be more realistic. The behavior of parameterized convection schemes to produce too often too light precipitation is consistent with several other studies (see introduction). We will clarify these points in a revised version of our manuscript.

*Review: L65ff, Sect. 2.1: It would be nice to have a little more background on the simulation setup. E.g., what is the update frequency of the lateral boundary data?*

**Reply:** In L92 we describe that lateral boundary conditions are updated every 6 hours. These lateral boundary conditions are also obtained from variable fields from the MPI-ESM Holocene simulations (L72-73).

*Review: L65ff, Sect. 2.1: I would also be interested in seeing how the ICON-NWP runs compare to the precipitation from the global MPI-ESM model. E.g., is 40km-P also better than MPI-ESM?*

**Reply:** A direct comparison between the MPI-ESM Holocene simulation and the 40km-P simulation is not possible at this stage of our study. The MPI-ESM uses a dynamic land-vegetation scheme (JSBACH), and therefore, it simulates an extended greening of the Holocene Sahara. In contrast, we have prescribed present-day conditions in the Sahara for the nested regional climate simulations. We did this as a first step – like it was done in the Paleo Modeling Intercomparison Project (PMIP) Phase 1. Hence, comparing the 40km-P simulation to the global MPI-ESM model would be an unfair comparison.
In a second set of simulations, we have prescribed a green Sahara, consistently with the MPI-ESM-Simulations, and we have done additional sensitivity studies using different soil moisture configuration to see whether the results obtained in the current study are affected by the land boundary conditions. Currently we analyze these data. We have found little change so far. Hence our results of the current study are qualitatively robust. A paper is in preparation in which we will certainly take up your suggestion.

*Review: L94ff: I would like to see a bit more justification for the chosen years, especially since later only one of the years is studied in more detail. E.g., a figure would help to make the arguments more explicit.*

**Reply:** After the first 15 years of the spinup simulation we choose two years: 1. based on the JJASO mean precipitation rate over land points over north Africa (37°W-52°E, 0°N-40°N) and 2. based on the northward propagation of precipitation. We looked for a combination of relatively high (weak) mean JJASO precipitation rate and strong (weak) northward extension for the strong monsoon year (weak monsoon year). We will add a figure and a short explanation in the revised manuscript.

*Review: L105: How is the diurnal cycle modified? And why is the 5km-E version also affected by this change (Fig. 2b)? It then seems the change cannot be a tuning parameter of the convection scheme.*

**Reply:** Peter Bechtold developed a new CAPE-based closure for the convection scheme, which is described in Bechtold et al. (2014). The new closure is not only based on CAPE but also takes into account boundary layer forcing. The boundary layer forcing is included via a boundary layer time scale that acts to delay the development of deep convection. Depending on the chosen time scale, the convection can be more or less delayed. If the boundary layer time scale is set to the deep convective adjustment time scale, then the boundary layer forcing is not taken into account, which leads to a more rapid development of convection with a midday peak.

The 5km-E version can be slightly affected because of the nesting setup we use. The 40km domain is the parent domain of the simulation. It drives (via boundary and initial conditions) the 20km domain, the 20km domain drives the 10km domain and the 10km finally drives the 5km domain. Therefore,

modifications in the 40km domain can yield small variations in the 5km domain. We will add this information in a revised version of our manuscript.

**Review:** *L125: I find the wording of "per latitude" unnecessary or confusing. The units of precip are mm/day and not mm/day/latitude.*
**Reply:** will be corrected

**Review:** *Fig. 3, caption: Domain a should probably read WAM domain.*
**Reply:** It should read WA-Domain what stands for "West Africa". We will spell out the acronym. We also clarify this in Fig. 1.

**Review:** *L188: I assume the local drying refers to the runoff described later. Maybe this can be hinted at already here so as to help orient the reader?*

**Reply:** Yes, the local drying refers to the drying due to the high runoff. We will make this clearer in the revised version.

**Review:** *Fig. 8: In addition to the maps it would be nice if you could calculate the moisture flux into/out of the WAM domain. It's a bit hard to see from the maps.*

**Reply:** Yes, we will calculated the domain mean moisture flux over the WA-Domain and add the num ber in the text in the revised version.

**Review:** *L249: I suggest you include a sentence of the end of this section that I assume should say that there is more moisture advection in the 5km run, so this cannot explain the drier atmosphere.*

**Reply:** Will be done. The maps and the calculation suggest that the moisture advection into the WA-domain is stronger in the 40km-P simulation than in the 5km-E simulation. This again confirms that the precipitation in the 40km-P simulation is higher than in the 5km-E simulation. Only in the Gulf of Guinea, the moisture transport is stronger in the 5km-E simulation. This suggests that the moisture advection from the Gulf of Guinea in the 5km-E simulation is either not large enough to overcompensate the drying induced by the runoff or/and the moisture is not transported sufficiently inland.

**Review:** *Data statement: I would like to see a proper data statement. Can the simulations be made public? I found the analysis scripts and the runs scripts in the linked data file. That should be described in more detail.*

**Reply:** The MPI good scientific practice only includes primary data. Primary data includes the model code and the needed input data files to re-run the simulations. The full simulations are not included. However all data are available upon request.

All typos will be corrected in the revised version of the manuscript.

**Anonymous Reviewer #2**

**Response to the anonymous review on the manuscript cp-2020-162 "Influence of the representation of convection on the mid-Holocene West African Monsoon" by Leonore Jungandreas et al.**

We thank the reviewer for carefully reading and commenting on our manuscript. In the following, we reply to the comments point by point.

**Review**: *Jungandreas et al. provide an evaluation of how the parameterized and explicitly resolved deep convection in the ICON climate model affect the meridional distribution of the West African Monsoon at the mid-Holocene by comparing the JAS features with 40km patameterized (40km-P) and 5km explicitly resolved (5km-E) convection. This manuscript is in general well written. I only have a few major comments and a series of minor ones listed below.*

*Main comments/questions:*

*1. Methods*

*--The years: I'm confused about the years you've chosen from the simulations. You run the spinup simulations for the period 7039 to 7010 BP. Are these years referred to the exact years BP, or the model years in the simulation? And why did you pick up two years within the spinup run? My understanding of "spinup" is to let a condition reach its equilibrium state. If you choose a model year from the spinup simulation, the condition, here the soil moisture, might not have reached it equilibrium state.*

**Reply:** First, the years refer to the exact years BP. Second, the spinup simulations runs for 30 years. The soil moisture reaches equilibrium after around 15 years (7024 BP). The years for the analysis are then chosen from the 15 years between 7025BP and 7010BP to ensure that the soil moisture is stable already, as suggested by the reviewer. We will clarify this point when revising our manuscript.

**Review:** *--The resolutions: Though you've used 10km-P and 10km-E simulations to show that the representation of convection is more important than the resolution, figure 3a also shows that the resolution does affect the precipitation rate in some way. I think it would be more convincing if you use the same resolution and only change the convection scheme. Or please explain/clarify your choice.*

**Reply:** The resolution does have an impact but we make the point that the choice of how the convection is parameterized has a bigger influence on the results. We also did the whole analysis with 10km explicitly resolved convection and 10km parameterized convection. We found that the mechanism, we describe in the paper, do not change compared to the 40km-P and the 5km-E simulation. We add the analysis in the Appendix of the paper.
Furthermore, the parametrization of convection is optimized to perform best at coarse resolutions, whereas explicitly resolved convection is best at finer horizontal resolution. Our idea was to use/compare the two resolutions where parameterized or explicitly resolved convection, respectively, perform best - especially after we investigated that the processes remain unchanged when we use 10km horizontal resolution.

**Review:** *2. Baselines*
*I would suggest to also give the baselines (and reconstructions if possible) in figures. Directly comparing 40km-P simulation with 5km-E simulation is not clear to show which one is better.*

**Reply:** Because we do not prescribe a vegetated Sahara yet, the comparison with any Proxies is difficult. However, we will revise our manuscript and carfully re-formulate sentences where we imply that one simulation is better than the other.

*Review: Minor comments:*
*Lines 31-33: The description of the comparison here is confusing. PMIP3&4 simulations simulate the annual mean precipitation anomalies over the Sahel region at about 300-400 mm/year on average.*

**Reply:** That's correct. We had the simulations from Pausata et al.,2016 and Egerer et al.,2018 in mind when we wrote this. We will correct this in the revised manuscript.

*Review: Lines 44-45: The resolution is not always that coarse. However, GCMs usually use a coarse resolution version to run paleo experiments.*

**Reply:** In a revised version, we will re-formulate this statement: For most, if not all, palaeo simulations GCMs are run at a rather coarse resolution at horizontal scales of more than 200 km.

*Review: Line 68: Change "hasalready" to "has already"*
**Reply:** will be corrected in revision

*Review: Figure 1: I would suggest to give the latitude and longitude boundaries of the coastal, Sahel and Sahara regions, respectively.*

**Reply:** We will add this information in the revised manuscript:
The latitude and longitude boundaries are:
Coastal region: 6°N-14°N, 5°W-10°E
Sahel region : 14°N-21°N, 5°W-10°E
Sahara region: 21°N-28°N, 5°W-10°E

*Review: Figure 2, 5, 9 captions: Fig1 doesn't have panel a, rewrite the corresponding words appropriately.*
**Reply:** will be corrected

*Review: Lines 126-127: There is no need to restart a new paragraph.*
**Reply:** We will removed it in the revised manuscript.

*Review: Line 195: Rewrite "Fig1: b)…d) the Saharan region" to match your figure 1.*
**Reply:** will be corrected.

**Response to the anonymous review on the manuscript cp-2020-162 "Influence of the representation of convection on the mid-Holocene West African Monsoon" by Jungandreas et al.**

We thank the reviewer for carefully reading and commenting on our manuscript. In the following, we reply to the comments point by point.

*Review:*
*Manuscript Summary*
*The authors examine the influence of parameterized/resolved convection and model resolution on the simulation of mid-Holocene African climate. The simulation with parameterized convection exhibits greater and more widespread (further north) rainfall over northern Africa. The authors point to an important role of soil moisture and evaporation in driving differences in the parameterized/resolved convection simulations. Specifically, they find that isolated, heavier rain events in the simulation with resolved convection enhance runoff, reducing the amount of water in the soil and ultimately reducing evaporation and moisture recycling.*

*The manuscript is organized and well written. The hypothesis that heavier rain events reduce the amount of moisture available for recycling is very interesting. I suggest the authors address the following comments to strengthen the manuscript.*
*Suggested Revisions*

*Review 1)*
*Your closing point in the Discussion about the potential influence of plants on the results is a major one. You should make this point earlier in the manuscript. Perhaps you could end Section 2.1 with a sentence or two about this. I also suggest highlighting the potentially important influence of proper vegetation-soil moisture interactions when comparing parameterized versus resolved-convection simulations in the Abstract.*

**Reply:** We will add this point in the revised manuscript.

*Review 2)*
*The mid-Holocene simulation does not exhibit precipitation amounts that match those estimated from proxies (the simulation is too dry). This should be noted, and the implications of this for the interpretation of your results should be discussed.*

**Reply:** We do not expect the precipitation amounts to match those estimated from proxies because we do not simulate with a vegetated Sahara yet. The land surface conditions reflect present-day conditions. We will make this clearer in the revised manuscript.

*Review 3)*
*At line 105 you note:*
*"To test the importance of the timing of the diurnal cycle for the representation of the monsoon propagation during the mid-Holocene, we perform a second set of nested simulations where we modify the timing of the diurnal cycle in the simulation with parametrization convection."*
*From that description, it doesn't appear you modified the diurnal cycle in the resolved convection simulation. However, in Figure 3b, the 5km-E simulation has different precipitation values from the 5km-E simulation in panel (a).*

**Reply:** That is true. The diurnal cycle in the 5km-E simulation is not changed but the variations evolve due to the nesting strategy we use. The 40km domain is the parent domain of the simulation setup. It drives (via boundary and initial conditions) the 20km domain, the 20km domain drives the 10km domain and the 10km finally drives the 5km domain. Therefore, modifications in the 40km domain can yield small variations in the 5km domain. We will add this information in a revised version of our manuscript.

***Review 4)***
*How is runoff calculated in Figure 9? At line 333-334 you mention that the runoff is "surface runoff". Is subsurface runoff not included? If you are currently only showing surface runoff, you should also include a figure of total runoff.*

**Reply:** In this version of the manuscript, we only presented the surface runoff. In the revised manuscript, we will show the total runoff that is the sum of surface runoff plus ground water runoff. The picture does not change qualitatively: the 5km-E simulation produces larger amounts of runoff than the 40km-P simulation.

***Review 5)***
*At lines260 and 332 you note:*
*"The lower soil moisture in the 5km-E simulation is due to higher amounts of surface runoff"*
*"The higher evapotranspiration rate is due to a higher soil moisture content throughout the whole simulation period. This is due to much weaker surface runoff in the 40 km-P than in the 5 km-E simulation".*
*I'm not sure you can attribute the soil moisture differences solely to the runoff differences. The greater precipitation in 40km-P likely contributes to greater soil moisture. Additionally, the heavier rainfall in the 5km-E simulation may promote more infiltration, and that may also explain why the soil moisture in the top layer is reduced in 5km-E. Does the soil moisture content for the entire soil column also differ between the simulations? I recommend showing this.*

**Reply:** Yes, the soil moisture content is larger in all soil layers in the 40km-P simulation than in the 5km-E simulation. We will replace the timeseries of the uppermost soil layer soil moisture with the sum over the upper 6 (of 8) soil layers in the revised manuscript. The lowest two soil layers are climatological layers.

***Review 6)***
*In lines 334 – 336, you note:*

*"In the 40 km-P simulation, light drizzle moistens the upper soil layers constantly, which makes it easier to trigger convection and to produce precipitation."*
*Can you provide some discussion about this? Is this true of all convective schemes or just the Tiedtke scheme? Has drizzle been shown to influence subsequent convection in the real world? A citation or two here would be helpful.*

**Reply:**
It was shown by several studies that parameterized convection tend to produce too much too light rainfall (i.e drizzle) as we pointed out in the introduction (L44-49). The studies cited used different GCMs, which suggests that it is a feature generated by different convective parameterization schemes.
What we want to point out with this sentence is that the light precipitation can be stored by the soil more easily without loosing much water into runoff. The soil stays moister as shown in the timeseries

of soil moisture in Fig. 9b. This is also true if we sum up all soil layers. The moister soil then favors higher evapotranspiration, moister atmospheric boundary layer, and stronger generation of convection.

**Review 7)**

*From Figure 5, it is apparent that the African Easterly Jet is weaker in the 5km-E simulation. This may have a few effects on precipitation in the simulation. One, a weaker AEJ should result in less moisture export from the continent (e.g., Cook 1999), keeping the atmosphere more humid in the Sahel in the 5km-E simulation. Two, it could result in weaker African Easterly Waves (AEW) through reduced baroclinic and barotropic energy conversions.  Though it is hard to say without doing an AEW analysis, it is possible that weaker AEWs result in fewer precipitation events in the 5km-E simulation. Cook, K. H. (1999). Generation of the African Easterly Jet and Its Role in Determining West African Precipitation, Journal of Climate, 12(5), 1165-1184.*

**Reply:** We will add this discussion point in the chapter in the revised version.

**Review 8)**

*In addition to the Skew-T diagrams in Figure 6, could you also present summer-average values of CIN, CAPE, etc. at 12UTC? These could be in a table.  It seems a bit arbitrary to select a single day to demonstrate the differences between the simulations.  By showing that these differences are consistent across the summer, your results will be more robust.*

**Reply:** That is true. We will add a table with the mean CAPE and CIN values for the JAS season that we analyzed in the paper.